# Learning Large Bayesian Networks with Expert Constraints

**Vaidyanathan Peruvemba Ramaswamy**[1]                    **Stefan Szeider**[1]

[1]Algorithms and Complexity Group, TU Wien, Vienna, Austria

## Abstract

We propose a new score-based algorithm for learning the structure of a Bayesian Network (BN). It is the first algorithm that simultaneously supports the requirements of (i) learning a BN of bounded treewidth, (ii) satisfying expert constraints, including positive and negative ancestry properties between nodes, and (iii) scaling up to BNs with several thousand nodes. The algorithm operates in two phases. In Phase 1, we utilize a modified version of an existing BN structure learning algorithm, modified to generate an initial Directed Acyclic Graph (DAG) that supports a portion of the given constraints. In Phase 2, we follow the BN-SLIM framework, introduced by Peruvemba Ramaswamy and Szeider (AAAI 2021). We improve the initial DAG by repeatedly running a MaxSAT solver on selected local parts. The MaxSAT encoding entails local versions of the expert constraints as hard constraints. We evaluate a prototype implementation of our algorithm on several standard benchmark sets. The encouraging results demonstrate the power and flexibility of the BN-SLIM framework. It boosts the score while increasing the number of satisfied expert constraints.

## 1 INTRODUCTION

Bayesian network structure learning is the computationally expensive problem of discovering a Bayesian network (BN) that optimally represents a given training data set [Chickering, 1996]. In addition to fitting the data, often measured in terms of a score function, several other requirements have been taken into account for the BN structure learning.

A fundamental requirement considered by an extensive volume of research is to learn BNs that fit the data and have *bounded treewidth* [Benjumeda et al., 2019, Berg et al.,

2014, Elidan and Gould, 2009, Nie et al., 2015, Scanagatta et al., 2016, 2018, Korhonen and Parviainen, 2013, Parviainen et al., 2014]. Bounded treewidth BNs admit tractable probabilistic inference [Kwisthout et al., 2010].

Another fundamental requirement receiving a growing amount of attention is to learn BNs that fit the data and satisfy additional *expert constraints* [Chen et al., 2016, Kennett et al., 2001, Li and van Beek, 2018, Corander et al., 2013]. Such constraints can assert, for instance, direct or indirect causation between random variables in terms of whether or not one variable is a parent or an ancestor of the other in the DAG of the learned BN. See Table 1 for a list of expert constraints considered in the literature.

Table 1: Various expert or side constraints considered in the literature. Here, *path* refers to simple directed paths.

|  |  |
|---|---|
| *Arc constraints (direct causation)* | |
| $u \rightarrow v$ | the DAG contains the arc $(u, v)$ |
| $u \nrightarrow v$ | the DAG does not contain the arc $(u, v)$ |
| $u \leftrightarrow v$ | the DAG contains either the arc $(u, v)$ or the arc $(v, u)$ |
| *Ancestry constraints (indirect causation)* | |
| $u \rightsquigarrow v$ | the DAG contains a path from $u$ to $v$ |
| $u \not\rightsquigarrow v$ | the DAG does not contain a path from $u$ to $v$ |
| $u \leftrightsquigarrow v$ | the DAG contains either a path from $u$ to $v$ or one in the other direction |

In addition to bounded treewidth and expert constraint requirements, one must address the *scalability* of methods for BN structure learning. For instance, learning a BN of bounded treewidth that optimally fits the data is NP-hard [Korhonen and Parviainen, 2013]. The consideration of expert constraints provides an additional source of complexity.

In this paper, we propose Con-BN-SLIM (Constrained BN-SLIM), the first method for BN structure learning that addresses all three requirements simultaneously: bounded

*Accepted for the 38th Conference on Uncertainty in Artificial Intelligence*  (UAI 2022).

Table 2: Feature Comparison Table. [†] CaMML allows weighted constraints with the weight of 1 signifying hard constraints. [‡] Negative constraints are treated as hard constraints, while positive constraints can be violated.

| | Scalability (# RVs) | Bounded treewidth | Supported constraints | Score optimization |
|---|---|---|---|---|
| EC Tree [Chen et al., 2016] | $\leq 20$ | no | $\{\rightsquigarrow, \not\rightsquigarrow\}$ | exact |
| MINOBSx [Li and van Beek, 2018] | $\leq 50$ | no | $\{\not\rightarrow, \not\rightsquigarrow, \rightarrow, \leftrightarrow, \rightsquigarrow\}$ | approx. |
| CaMML [Kennett et al., 2001] | unknown | no | $\{\not\rightsquigarrow, \rightarrow, \leftrightarrow, \rightsquigarrow\}^{\dagger}$ | exact |
| k-greedy [Scanagatta et al., 2018] | $\leq 10000$ | yes | $\varnothing$ | approx. |
| BN-SLIM [PR and Szeider, 2021a] | $\leq 10000$ | yes | $\varnothing$ | approx. |
| Con-k-greedy (this paper) | $\leq 10000$ | yes | $\{\not\rightarrow, \not\rightsquigarrow, \rightarrow, \leftrightarrow, \rightsquigarrow\}^{\ddagger}$ | approx. |
| Con-BN-SLIM (this paper) | $\leq 10000$ | yes | $\{\not\rightarrow, \not\rightsquigarrow, \rightarrow, \leftrightarrow, \rightsquigarrow\}^{\ddagger}$ | approx. |

treewidth, expert constraints, and scalability. Table 2 shows how our new method compares to other BN structure learning methods from the literature. Since these methods span decades of research, it was a natural choice to try and reuse their progress as much as possible so as to stand on the shoulders of giants. Thus, we arrived at our 2-phase approach of Con-BN-SLIM, which leverages the scalability of k-greedy and the localized optimization power of BN-SLIM (particularly useful for expert constraints).

In Phase 1, a heuristic algorithm greedily computes a candidate BN from data, thereby trying to satisfy as many expert constraints as possible. The heuristic algorithm is a version of the k-greedy algorithm by Scanagatta et al. [2018] that we modified to consider expert constraints. This method scales very well. However, considering expert constraints significantly deteriorates the algorithm's capability of fitting the BN to the data. This even prevails when we consider the expert constraints as soft constraints, which allows the algorithm to violate some constraints.

We, therefore, add a Phase 2 that takes the candidate BN from the first phase and repeatedly tries to improve the score by optimizing local parts of the BN. The second phase is an extension of the BN-SLIM approach by Peruvemba Ramaswamy and Szeider [2021a]. BN-SLIM utilizes a MaxSAT solver to locally improve the BN. Crucial for our extension is to express suitable local versions of the desired expert constraints in terms of hard constraints for the MaxSAT solver. This way, the solver may improve the fitting of the BN while maintaining the satisfaction of all the expert constraints satisfied by the first phase solution.

Due to our novel contributions in Section 4, like localization of global constraints and the scaffolding of auxiliary variables required to express and incorporate expert constraints into BN-SLIM, the proposed approach is more than just gluing together existing methods.

We evaluated a prototype implementation of Con-BN-SLIM on all discrete sample data from the bnlearn BN repository, sampling expert constraints from the ground truth networks. After the first phase of running the modified heuristic algo-

rithm for about 30 minutes, the rate of improvement deteriorates. Phase 2 begins, and Con-BN-SLIM takes over the candidate network and shows a remarkably high improvement rate. The final network shows a significantly higher score than the one produced by Phase 1, which displays favorably in the $\Delta$BIC metric.

The empirical findings on our prototype implementation are highly encouraging, providing the ground for several avenues of further investigation.

## 2 PRELIMINARIES

In this section, we provide a brief overview of the required background. Throughout this section, we closely follow the general notation and methodology of Peruvemba Ramaswamy and Szeider [2021a] From this point on, we use the shorthand *heuristic* to refer to heuristic algorithms, i.e., algorithms that do not guarantee their solution's optimality.

### 2.1 STRUCTURE LEARNING

We consider the problem of learning the structure (i.e., the DAG) of a BN from complete data set of $N$ instances $D_1, \ldots, D_N$ over a set of $n$ categorical random variables $X_1, \ldots, X_n$. The goal is to find a DAG $D = (V, E)$ where $V$ is the set of nodes (one for each random variable) and $E$ is the set of arcs (directed edges) as 2-tuples. The value of a *score function* determines how well a DAG $D$ fits the data; the DAG $D$, together with local parameters (i.e., conditional probabilities), forms the BN Koller and Friedman [2009].

We assume that the score is *decomposable*, i.e., being constituted by the sum of the individual random variables' scores. Hence, we can assume that the score is given in terms of a *score function* $f$ that assigns each node $v \in V$ and each subset $P \subseteq V \setminus \{v\}$ a real number $f(v, P)$, the *score* of $P$ for $v$. The score of the entire DAG $D = (V, E)$ is then $f(D) := \sum_{v \in V} f(v, P_D(v))$ where $P_D(v) = \{u \in V : (u, v) \in E\}$ denotes the *parent set* of $v$ in $D$. This set-

ting accommodates several popular scores like AIC, BDeu, and BIC Akaike [1974], Heckerman et al. [1995], Schwarz [1978]. If $P$ and $P'$ are two potential parent sets of a random variable $v$ such that $P \subsetneq P'$ and $f(v, P') \leq f(v, P)$, then we can safely disregard the potential parent set $P'$ of $v$. Consequently, we can disregard all nonempty potential parent sets of $v$ with a score $\leq f(v, \emptyset)$. Such a restricted score function is a *score function cache*.

## 2.2 TREEWIDTH

A *tree decomposition* $\mathcal{T}$ of a graph $G$ is a pair $(T, \chi)$, where $T$ is a tree and $\chi$ is a function that assigns each tree node $t$ a set $\chi(t)$ of vertices of $G$ such that the following conditions hold:

**T1** For every edge $(u, v)$ of $G$ there is a tree node $t$ such that both $u, v \in \chi(t)$.

**T2** For every vertex $v$ of $G$, the set of tree nodes $t$ with $v \in \chi(t)$ induces a non-empty subtree of $T$.

The sets $\chi(t)$ are called *bags* of the decomposition $\mathcal{T}$, and $\chi(t)$ is the bag associated with the tree node $t$. The *width* of a tree decomposition $(T, \chi)$ is the size of a largest bag minus 1. The *treewidth* of $G$, denoted by $\mathrm{tw}(G)$, is the minimum width over all tree decompositions of $G$.

The *treewidth-bounded BN structure learning problem* takes as input a set $V$ of nodes, a decomposable score function $f$ on $V$, and an integer $W$, and it asks to compute a DAG $D = (V, E)$ whose moral graph has treewidth $\leq W$, such that $f(D)$ is maximal. The moral graph of a DAG $D$ is obtained by treating all arcs as undirected and inserting arcs between two nodes if they share a common child.

## 2.3 EXPERT CONSTRAINTS

In our work, we consider only arc and ancestry constraints. The requirements for satisfaction of the constraints is described in Table 1. We use the term *constraint set* to refer to a set of such constraints and a DAG $D$ is said to satisfy a constraint set if it satisfies all constituent constraints. We refer to $\rightarrow$ and $\rightsquigarrow$ as *positive* constraints and $\not\rightarrow$ and $\not\rightsquigarrow$ as *negative* constraints. Note that, $u \not\rightsquigarrow v$ is denoted as $v > u$ by Li and van Beek [2018]. Also note that, some other variants of constraints like $\not\leftrightsquigarrow$ can be expressed as boolean combinations of the elementary constraints from Table 1.

Given a DAG over a set $V$ of vertices, a *path* $P$ is a sequence $v_1, \ldots, v_\ell$ of vertices such that there exists an arc from $v_i$ to $v_{i+1}$ for all $1 \leq i < \ell$. Since the graph is acyclic, all involved $v_i$ are distinct. The path $P$ *avoids* a set $S \subseteq V$ if $v_i \notin S$ for all $1 \leq i < \ell$.

Finally, we also use the concept of partial orders in our modification of k-greedy (Section 3). A partial order is a

set of pairwise ordering requirements $u \triangleright v$. A linear order $u_1, \ldots, u_n$ is said to *obey* a partial order if, for every $u_i \triangleright u_j$ in the partial order, $i < j$.

## 3 K-GREEDY WITH CONSTRAINTS

In this section, we describe the modifications made to k-greedy to obtain a heuristic algorithm to solve the Constrained BN structure learning problem. We would like to point out that we chose to modify k-greedy as a proof of concept because of its simplicity. However, theoretically similar modifications are also possible for the more aggressive k-MAX heuristic [Scanagatta et al., 2018].

### 3.1 OVERVIEW OF K-GREEDY

First, we briefly overview the basic k-greedy heuristic by Scanagatta et al. [2016]. The algorithm takes as input a set $X$ of RVs and a score function cache and returns a DAG $D$ along with a corresponding (rooted) tree decomposition $T$. The algorithm repeatedly performs the following steps:

Step 1. Randomly sample a linear ordering $\sigma$ over the variables $X$

Step 2. Construct the root bag of $T$ from the first $k + 1$ variables of $\sigma$. Also, compute a DAG over these variables maximizing the score (either exactly or approximately).

Step 3. Then insert the remaining variables from $\sigma$ one by one into the DAG, selecting the best parent set for it from the already inserted variables.

After each step, if the newly computed DAG has a higher score than the previous best DAG, it is called an *improvement*.

### 3.2 MODIFIED K-GREEDY

To upgrade this algorithm to work with expert constraints, we modify each of the steps above to obtain Con-k-greedy (Constrained k-greedy). Algorithm 2 shows the pseudocode for Con-k-greedy. In Step 1, instead of randomly sampling an order, we first 'compile' the supplied constraints $\mathcal{C}$ into a partial order $\mathcal{P}$. Meaning, we add a partial order pair $u \triangleright v$ to $\mathcal{P}$ for every positive constraint $u \bowtie v$, i.e., $\bowtie \in \{\rightarrow, \rightsquigarrow\}$. This is because it can be easily shown that all topological orderings of all networks that satisfy constraints $\mathcal{C}$ also obey the partial order $\mathcal{P}$. We then randomly sample linear orderings that obey this partial order, which serve as both elimination orderings and topological orderings for the DAG being constructed.

In Step 2, we now search for a best DAG that does not violate any negative constraint by brute force or using any other

```
Input   : Set C of expert constraints
Output  : Set P of partial order pairs
begin
    P ⟵ ∅
    foreach u ⋈ v ∈ C do
        if ⋈ ∈ {→, ⤳} then
            P ⟵ P ∪ {u ▷ v}
        end
    end
    return P
end
```

**Algorithm 1:** Pseudocode for `Compile`

local solver (Line 1). In Step 3, we select the best parent set from among the parent sets that violates none of the negative constraints and satisfies all the positive constraints involving the currently inserted variable. If there are no such parent sets, we simply select the empty parent set for the current variable (see Line 2); this ensures that no negative constraints are violated.

This results in an algorithm that can keep generating better and better scoring DAGs with the condition that all generated DAGs respect the negative constraints from C as hard constraints and the positive constraints as soft constraints.

### 3.3 PRACTICAL CONSIDERATIONS

Theoretically, it is possible to modify k-greedy similarly so that the resultant algorithm treats all constraints as hard constraints. However, in practice, we noticed that this severely limits the number of improvements and, in many cases, fails to find any networks. We, thus, slightly alter Step 3 to only reject choices of parent sets that violate the negative constraints, i.e., $\{\not\rightsquigarrow, \not\rightarrow\}$. As a result, the heuristic provides solutions which satisfy all the negative constraints but not necessarily all the positive and undirected constraints. In other words, all positive and undirected constraints, i.e., $\{\rightarrow, \leftrightarrow, \rightsquigarrow\}$, are treated as *soft* constraints.

### 3.4 EXPERIMENTS

We experimentally evaluated the heuristic proposed above and found the results unsatisfactory. We noticed that the rate of improvement diminishes quite quickly and essentially reaches saturation by 30 mins (see Figure 1). However, the output of the heuristic could serve as a starting point for further improvement. The *SAT-based Local Improvement Method* (SLIM) framework was introduced by Lodha et al. [2016, 2019] and later used by Fichte et al. [2017], Peruvemba Ramaswamy and Szeider [2020, 2021a,b], Schidler and Szeider [2021] could potentially turbocharge and improve the score of such an intermediate saturated solution. In the next sections, we develop a solution using the SLIM

```
Input   : Score function f, set C of expert
            constraints, treewidth bound k
Output  : DAG D satisfying all negative constraints
            necessarily and positive constraints
            optionally
begin
    P ⟵ Compile(C)
    loop
        Sample linear order σ obeying P
        Construct root bag B₀ ⟵ {σ₀, ..., σ_{k+1}}
1:      Construct a DAG D over B₀ maximizing
        score and not violating any negative
        constraints
        for v in σ_{k+2}, ..., σ_n do
            R ⟵ set of parent sets of v not
            violating any negative constraints and
            satisfying all positive constraints of the
            form u ⋈ v for some u
            if R is nonempty then
                P_D(v) ⟵ maximum score parent
                set from R
            else
2:              P_D(v) ⟵ ∅
            end
        end
        if algorithm terminated then
            return D
        end
    end
end
```

**Algorithm 2:** Pseudocode for Con-k-greedy

framework for the Constrained BN structure learning problem.

## 4  BN-SLIM WITH CONSTRAINTS

### 4.1  THEORY

In this section, we lay the theoretical foundation for solving the Constrained BN structure learning problem using the SLIM framework. The SLIM framework has been previously used by Peruvemba Ramaswamy and Szeider [2021a] to solve the BN structure learning problem. We refer to this method as BN-SLIM. We directly extend BN-SLIM to solve the Constrained BN structure learning problem; as a result we reuse the same notation.

The problem input consists of a set $V$ of random variables, a score function $f$, a treewidth bound $W$ and a set of expert constraints $C$. We allow $C$ to contain constraints of type $\{\rightarrow, \not\rightarrow, \leftrightarrow, \rightsquigarrow, \not\rightsquigarrow\}$.

The goal is to compute a DAG $D^\star$ over $V$ with maximum

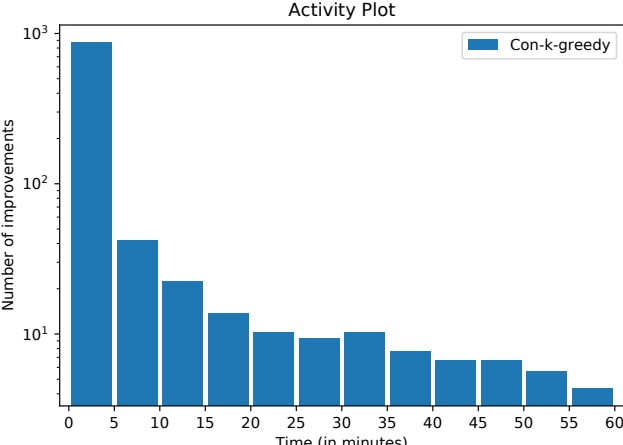

Figure 1: Activity plot showing the rate of improvements of Con-k-greedy against time. Note that the y-axis is in logscale.

score such that the treewidth of the moralized graph $M(D^\star)$ is bounded by $W$ and $D^\star$ satisfies all the constraints in $\mathcal{C}$. We assume to have an initial heuristic solution $D$, a corresponding tree decomposition $\mathcal{T} = (T, \chi)$ of width $\leq W$ of the moralized graph $M(D)$ and that $D$ satisfies the constraint set $\mathcal{C}$. Our aim now, is to compute a DAG $D^{\text{new}}$ over $V$ with score at least as much as $D$ while still having bounded treewidth and satisfying constraint set $\mathcal{C}$. Applying this process repeatedly, we can improve the score of the resultant DAG while still satisfying all the requirements.

We select a subtree $S \subseteq T$ such that the total number of vertices in $V_S := \bigcup_{t \in S} \chi(t)$ is at most some *budget* $B$ (a fixed constant limiting the size of the local instances such that instances of this size can be solved reasonably quickly by the local solver). The value of $B$ is decided by means of experimenting and educated guesses. We define $D_S^{\text{new}}$ as the DAG induced by $D^{\text{new}}$ on $V_S$, where $E(D_S^{\text{new}}) = \{ (u, v) \in E(D^{\text{new}}) : \{u, v\} \subseteq V_S \}$ and $\mathcal{S}^{\text{new}} = (S^{\text{new}}, \chi^{\text{new}})$ as a tree decomposition of $D_S^{\text{new}}$. For convenience, we use the shorthand $E_S^{\text{new}}$ to denote $E(D_S^{\text{new}})$.

We distinguish between different kinds of vertices:

- $v \in V_S$ a *boundary vertex* if there exists a tree node $t \in V(T) \setminus V(S)$ such that $v \in \chi(t)$;
- $v \in V_S$ is an *internal vertex* if $v$ is not a boundary vertex;
- $v \in V \setminus V_S$ is an *external vertex*.

Two boundary vertices $v, v'$ are *adjacent* if both occur together in some bag outside $S$. In that case we call $\{v, v'\}$ a *virtual edge*. We let $E_{\text{virt}}$ be the set of all virtual edges. The *extended moral graph* $M_{\text{ext}}$ is obtained from $M(D_S^{\text{new}})$ by adding all virtual edges. If $v, v'$ are two adjacent boundary vertices such that $D^{\text{new}}$ contains a directed path from $v'$ to $v$, where all the vertices on the path, except for $v'$ and $v$, are

external, then $(v', v)$ is a *virtual arc*. $E_{\text{virt}}^{\rightarrow}$ denotes the set of all virtual arcs.

We can now reiterate the conditions from Peruvemba Ramaswamy and Szeider [2021a] needed to state the main theorem.

**C1** $D_S^{\text{new}}$ is acyclic.

**C2** The moral graph $M(D_S^{\text{new}})$ has treewidth $\leq W$.

**C3** $\mathcal{S}^{\text{new}}$ is a tree decomposition of the extended moral graph $M_{\text{ext}}$.

**C4** For each $v \in V_S$, if $P_{D^{\text{new}}}(v)$ contains external vertices, then there is some $t \in V(T) \setminus V(S)$ such that $P_{D^{\text{new}}}(v) \cup \{v\} \subseteq \chi(t)$.

**C5** The digraph $(V_S, E_S^{\text{new}} \cup E_{\text{virt}}^{\rightarrow})$ is acyclic.

**Theorem 1** ([Peruvemba Ramaswamy and Szeider, 2021a]). *If all the conditions C1–C5 are satisfied, then $D^{\text{new}}$ is acyclic, the treewidth of $M(D^{\text{new}})$ is at most $W$, and the score of $D^{\text{new}}$ is at least the score of $D$.*

We now discuss how the different types of constraints can be transformed into their respective local versions along with the correctness for the same. We note that the input constraint set can only consist of elementary arc and ancestry constraints (listed in Table 1); however, the translation into their respective local versions additionally allows disjunctions over elementary constraints. This is because the local versions of the constraints are directly handed off to the MaxSAT solver which is capable of handling such disjunctions.

To discuss the behavior of the ancestry constraints, we use the concept of first-hit descendants and first-hit ancestors. Given a DAG $F$ over vertices $W$, subset $Y \subsetneq W$ and vertex $r \in W$, a node $s \in Y$ is said to be a first-hit descendant of $r$ in $Y$ if there exists a directed path from $r$ to $s$ avoiding all the other vertices in $Y \setminus \{r, s\}$. We denote by $\text{desc}_Y^r \subseteq Y$, the set of all *first-hit descendants of $r$ in $Y$*. Similarly, a node $s \in Y$ is said to be a first-hit ancestor of $r$ in $Y$ if there exists a directed path from $s$ to $r$ avoiding all the other vertices in $Y \setminus \{r, s\}$. We denote by $\text{anc}_Y^r \subseteq Y$, the set of all *first-hit ancestors of $r$ in $Y$*. We denote by $\top$, the always-true trivial constraint that is always satisfied.

**Arc constraints** $(\rightarrow, \nrightarrow, \leftrightarrow)$   Let $c$ be a constraint $u \bowtie v$, where $\bowtie \in \{\rightarrow, \nrightarrow, \leftrightarrow\}$. If either of $u, v \notin S$, then the constraint remains satisfied, since the presence or absence of the arc between $u, v$ is not affected by $D^{\text{new}}$. The local version of such a constraint is thus $\top$. Alternatively, if both $u, v \in S$, it suffices to ensure that the constraint $c$ holds in $D_S^{\text{new}}$. The local version of such a constraint is $c$ itself.

**Positive ancestry constraints** $(\rightsquigarrow)$   Consider a constraint of the form $u \rightsquigarrow v$. Since the constraint is satisfied in $D$, we know that there exists a $u - v$ path in $D$.

**Case 1** There is at least one $u - v$ path avoiding $V_S$. The constraint remains satisfied independent of $D_S^{\text{new}}$. The local version of such a constraint is $\top$.

**Case 2** All $u - v$ paths pass through $V_S$. It suffices to ensure that there exists at least one path in $D_S^{\text{new}}$ from some $d_u \in \text{desc}_{V_S}^u$ to some $a_v \in \text{anc}_{V_S}^v$. The local version of such a constraint is $\bigvee_{d_u, a_v} d_u \rightsquigarrow a_v$.

**Negative ancestry constraints** ($\not\rightsquigarrow$) Consider a constraint of the form $u \not\rightsquigarrow v$. Since the constraint is satisfied in $D$, we know that there are no $u - v$ paths in $D$. Any $u - v$ path passing through $V_S$ must be of the form $u - d_u - a_v - v$, for some $d_u \in \text{desc}_{V_S}^u, a_v \in \text{anc}_{V_S}^v$.

**Case 1** $\text{desc}_{V_S}^u = \emptyset$ or $\text{anc}_{V_S}^v = \emptyset$. The constraint remains satisfied independent of $D_S^{\text{new}}$. The local version of such a constraint is $\top$.

**Case 2** Both sets are non-empty. It suffices to ensure that there is no path in $D_S^{\text{new}}$ from any $d_u \in \text{desc}_{V_S}^u$ to any $a_v \in \text{anc}_{V_S}^v$. The local version of such a constraint is $\bigwedge_{d_u, a_v} d_u \not\rightsquigarrow a_v$.

From this discussion, we can assert the following lemma.

**Lemma 1.** *If $D_S^{\text{new}}$ satisfies the local versions of each of the constraint in $\mathcal{C}$, then $D^{\text{new}}$ satisfies the constraint set $\mathcal{C}$.* $\square$

From Theorem 1 and Lemma 1, we obtain the following corollary.

**Corollary 1.** *If conditions C1–C5 are satisfied and $D_S^{\text{new}}$ satisfies the local versions of the constraints in $\mathcal{C}$, then $D^{\text{new}}$ is acyclic, the treewidth of $M(D^{\text{new}})$ is at most $W$, the score of $D^{\text{new}}$ is at least that of $D$ and $D^{\text{new}}$ satisfies the constraint set $\mathcal{C}$.*

## 4.2 ENCODING

In this section, we describe the MaxSAT encoding to compute $D_S^{\text{new}}$. We build on top of the encoding by Peruvemba Ramaswamy and Szeider [2021a]. Briefly, the basic variables in the encoding are the $\text{par}_v^P$ variables, which are true if and only if $P$ is the parent set of $v$. These variables appear in the encoding as soft clauses weighted by $f(v, P)$. In addition to that, there are several hard clauses involving $\text{arc}_{u,v}$, $\text{acyc}_{u,v}$ and $\text{ord}_{u,v}$ variables, which encode the edges of the moralized graph, the acyclicity of the DAG and the elimination ordering corresponding to the tree decomposition respectively. The soft and hard clauses of the encoding are passed to the MaxSAT solver to optimize the network's score. The MaxSAT solver then finds solutions satisfying all the hard clauses while also maximizing the weight of the satisfied soft clauses. Eventually, this encoding finds a network with maximum score that satisfies the conditions C1–C5. For the sake of brevity, we skip repeating the entire encoding and only describe the additions.

Now, having Corollary 1, we describe the addition to the encoding that ensures that $D_S^{\text{new}}$ satisfies the local versions of the constraints.

**Arc constraints** ($\rightarrow, \nrightarrow, \leftrightarrow$) We filter out the infeasible parent sets based on the arc constraints. More specifically, for the constraint $u \rightarrow v$, we discard all parent sets of $v$ that do not contain $u$, and conversely, for the constraint $u \nrightarrow v$, we discard all parent sets that contain $u$.

**Ancestry constraints** ($\rightsquigarrow, \not\rightsquigarrow$) We address the ancestry constraints by introducing the following variables to keep track of the paths within the network:

1. $\text{dagarc}_{u,v}$ represents an arc in the DAG from $u$ to $v$ (does not include the moralized and fill-in edges unlike $\text{arc}_{u,v}$),

2. $\text{tarc}_{u,v}$ captures the transitive closure of the $\text{dagarc}_{u,v}$ variables.

3. $\text{path}_{u,v}$ implies the existence of a path in the DAG from $u$ to $v$,

4. $\text{pathq}_{u,v,z}$ is a helper variable for $\text{path}_{u,v}$ and implies the existence of a path in the DAG from $u$ to $v$ with $z$ as the penultimate vertex,

5. $\text{virtarc}_{u,v}$ represents the short-circuited directed paths through nodes outside the local instance.

We then introduce hard clauses over these variables to capture their semantics and to allow expressing expert constraints. This implies that these constraints are treated as hard constraints. At times, we write the clauses using the friendlier implication notation. However, all of these clauses can be converted into the standard *Conjunctive Normal Form* (CNF) required by the MaxSAT solver. For this reason, the encoding accepts as input all the elementary constraints as well as disjunctions over elementary constraints.

Phase 2 only considers the set of constraints satisfied by the initial heuristic solution as hard constraints. This ensures that all the constraints satisfied by the Phase 1 solution remain satisfied at the end of Phase 2. Further, there might be some constraints that were previously violated in the heuristic solution but end up being coincidentally satisfied by Phase 2. Thus, the set of satisfied constraints by the Phase 2 solutions is a (not necessarily strict) superset of the set of constraints satisfied by the Phase 1 solution.

To disallow simultaneous arcs in opposite directions in the DAG, we add the clauses

$$\neg\text{dagarc}_{u,v} \vee \neg\text{dagarc}_{v,u} \qquad \text{for all } u \neq v \in S.$$

We then add the following clauses to ensure that $\text{dagarc}_{u,v}$ is true if and only if $u$ is in the parent set of $v$.

$$\text{par}_v^P \Rightarrow \bigwedge_{u \in P} \text{dagarc}_{u,v} \qquad \text{for all } v \in S, P \in \mathcal{P}_v, \text{ and}$$

$$\text{dagarc}_{u,v} \Rightarrow \bigvee_{P \in \mathcal{P}_v \text{ s.t. } u \in P} \text{par}_v^P \quad \text{for all } u \neq v \in S.$$

And finally, we propagate the DAG arcs to the arcs of the moralized graph using the clauses

$$\text{dagarc}_{u,v} \Rightarrow \text{arc}_{u,v} \qquad \text{for all } u \neq v \in S.$$

For the $\text{tarc}_{u,v}$ variables, we initialize the transitivity using the $\text{dagarc}_{u,v}$ and $\text{virtarc}_{u,v}$ variables as follows

$$\left.\begin{array}{r} \text{dagarc}_{u,v} \Rightarrow \text{tarc}_{u,v} \\ \text{virtarc}_{u,v} \Rightarrow \text{tarc}_{u,v} \end{array}\right\} \quad \text{for all } u \neq v \in S,$$

and then encode the transitivity using the following clauses

$$\text{tarc}_{u,v} \wedge \text{tarc}_{v,w} \Rightarrow \text{tarc}_{u,w} \quad \text{for all distinct } u, v, w \in S.$$

To encode the path variables, we first encode the condition that the path can either be a single arc in the DAG, a single external virtual arc or a path through at least one other variable $z$. For this we add the following clauses for all $u \neq v \in S$,

$$\text{path}_{u,v} \Rightarrow \text{dagarc}_{u,v} \vee \text{virtarc}_{u,v} \vee \bigvee_{z \neq u,v} \text{pathq}_{u,v,z}.$$

Then, we encode the condition for the existence of a path from $u$ to $v$ with $z$ in the penultimate position, by asserting a path from $u$ to $z$ and either a direct arc or a virtual arc from $z$ to $v$. For this we add the following clauses for all distinct $u, v, z \in S$,

$$\text{pathq}_{u,v,z} \Rightarrow \text{path}_{u,z} \wedge (\text{dagarc}_{z,v} \vee \text{virtarc}_{z,v}).$$

Finally, we encode the constraints using the predicates described so far. For the arc constraints, we use the $\text{dagarc}_{u,v}$ variables as follows

$$\text{for } u \rightarrow v, \text{ we use } \text{dagarc}_{u,v},$$
$$\text{for } u \nrightarrow v, \text{ we use } \neg\text{dagarc}_{u,v},$$
$$\text{for } u \leftrightarrow v, \text{ we use } \text{dagarc}_{u,v} \vee \text{dagarc}_{v,u}.$$

For the ancestry constraints, we use the $\text{path}_{u,v}$ and $\text{tarc}_{u,v}$ variables as follows

$$\text{for } u \rightsquigarrow v, \text{ we use } \text{path}_{u,v},$$
$$\text{for } u \not\rightsquigarrow v, \text{ we use } \neg\text{tarc}_{u,v},$$

It is subtle but worth noting nonetheless, that the clause $\neg\text{path}_{u,v}$ does not ensure the absence of a path from $u$ to $v$ in the DAG, i.e., the $\text{path}_{u,v}$ variables can only be used to assert the existence of paths ($\rightsquigarrow$ constraints), not their absence ($\not\rightsquigarrow$ constraints). Which is why we use the $\text{tarc}_{u,v}$ variables to be able to assert the absence of paths.

# 5 EXPERIMENTS

## 5.1 SETUP

We tested the two proposed heuristics on a 4-core Intel Xeon E5540 2.53 GHz CPU cluster, with each process having access to 8 GB RAM. The k-greedy algorithm is available as a part of the BLIP package [Scanagatta, 2015] implemented in Java. We provide the relevant source code as a public GitHub repository[1]. We implemented Con-BN-SLIM by extending the publicly available BN-SLIM software [Peruvemba Ramaswamy and Szeider, 2021c], which uses the Python NetworkX library [Hagberg et al., 2008], and the UWrMaxSat[2] as the MaxSAT solver.

We ran the heuristics on score function caches and constraints sets generated from all the discrete networks available as a part of the bnlearn BN repository.[3] This repository is commonly used for benchmarking Bayesian Networks [Li and van Beek, 2018, Chen et al., 2016, Scanagatta et al., 2018, 2016]. We split up the networks into three groups—small, medium, and large—based on the number of random variables. We then synthesized expert constraints by randomly sampling a fixed number $\eta$ of constraints of each of the 5 types from the ground truth networks (see Table 3). Note that this repository consists of the networks themselves, not the instances or samples drawn from the BNs. Additionally, we also precomputed the treewidths of all the ground truth networks (ranging between 3 and 15) and used those values as the bounds for all the heuristics.

Table 3: Input Datasets

| Group | Variables | $\eta$ |
|-------|-----------|--------|
| Small | up to 50 | $\{5, 10\}$ |
| Medium | 50 to 500 | $\{10, 25, 50\}$ |
| Large | above 500 | $\{25, 50, 75\}$ |

## 5.2 METHOD

We now explain the format of the experiments used to compare the proposed heuristics, which is similar to that of Peruvemba Ramaswamy and Szeider [2021a]. We precompute the score function caches using the available functionality from the BLIP package. All the evaluated methods are supplied with the same score function caches. We then randomly synthesized different constraints using three random seed values. The score function caches along with a corresponding constraint set are together considered to be one

[1] https://github.com/aditya95sriram/bn-slim

[2] https://maxsat-evaluations.github.io/2019/descriptions.html

[3] https://www.bnlearn.com/bnrepository/

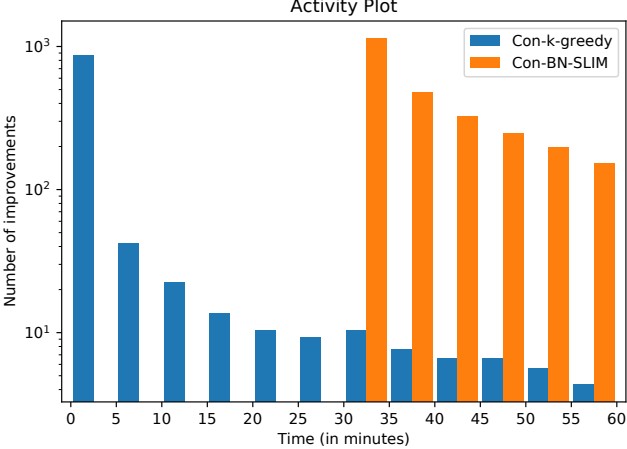

Figure 2: Activity plot showing rate of improvement of Con-BN-SLIM and Con-k-greedy against time. Note that the y-axis is in logscale.

input instance. This results in a total of 183 input instances.

We then ran the original k-greedy algorithm and the Con-k-greedy algorithm on these inputs for 60 minutes. For each input, we ran the heuristics with three different random seed values. For evaluating Con-BN-SLIM, we used the intermediate solution produced by Con-k-greedy at the 30-minute mark as the starting heuristic. After which, we run Con-BN-SLIM for another 30 minutes, thereby fixing the total runtime of each method to 60 minutes. For each input, we ran Con-BN-SLIM with 8 different configurations (random seed, timeout, encoding type). For all the experiments, we record the final score, the final satisfied constraint count, and the rate of improvement.

### 5.3 RESULTS

As a continuation to Figure 1, we first visualize the activity of Con-BN-SLIM compared to Con-k-greedy. Note that, Con-BN-SLIM only starts running at the 30-minute mark (after being handed the heuristic solution from Con-k-greedy) and hence does not record any improvements till that point. As is evident from Figure 2, despite the rate of improvements of Con-k-greedy slowing down drastically, when Con-BN-SLIM takes over, it is still able to find many improvements over the exact same networks. This demonstrates the notion of turbocharging quite well.

Next, we compare the scores of the networks produced by Con-k-greedy and Con-BN-SLIM at the 60-minute mark. We use the $\Delta$BIC metric to make this comparison. The difference in BIC scores of two networks approximates the ratio of their marginal likelihoods, which is the Bayes Factor [Raftery, 1995, Scanagatta et al., 2018]. The $\Delta$BIC score of a pair of networks is mapped to a categorical scale, with positive scores signifying positive evidence towards the

first network and vice versa. As can be seen from Table 4, Con-BN-SLIM severely outperforms Con-k-greedy.

Table 4: $\Delta$BIC values comparing Con-BN-SLIM against Con-k-greedy

| Category | $\Delta$ BIC | Count |
|---|---|---|
| extremely positive | $(10, \infty)$ | 127 |
| strongly positive | $(6, 10)$ | 0 |
| positive | $(2, 6)$ | 0 |
| neutral | $(-2, 2)$ | 14 |
| negative | $(-6, -2)$ | 1 |
| strongly negative | $(-10, -6)$ | 0 |
| extremely negative | $(-\infty, -10)$ | 7 |

Finally, we compare the constraint satisfaction by the solutions of k-greedy, Con-k-greedy, and Con-BN-SLIM in Table 5. We measure and tabulate the percentage of total constraints satisfied. There are several noteworthy points here.

Table 5: Comparison of Constraint Satisfaction as a Percentage of Total Constraints

| | Avg. % satisfied constraints | | |
|---|---|---|---|
| Group | k-greedy | Con-k-greedy | Con-BN-SLIM |
| Small | 77.74% | 84.52% | 90.24% |
| Medium | 63.80% | 74.43% | 81.73% |
| Large | 59.44% | 88.91% | 89.44% |
| All | 67.54% | 81.73% | 86.53% |

**k-greedy** We see that k-greedy, despite having no knowledge of the constraints, manages to satisfy more than half of them. This could be attributed to the fact that k-greedy still has access to the score function caches whose job is to quantify and reflect the closeness of any network to the ground truth network (just like the expert constraints).

**Con-k-greedy** We see a clear improvement in the constraint satisfaction by Con-k-greedy compared to k-greedy. This is to be expected as we modified the heuristic to consider the expert constraints.

**Con-BN-SLIM** We see that Con-BN-SLIM ends up satisfying slightly more constraints than Con-k-greedy even though it was not intentionally designed to do so. This, however, is a favorable side effect. Con-BN-SLIM never violates a constraint that was satisfied by the initial heuristic solution. Thus, by random chance, the number of satisfied constraints can only increase.

# 6 CONCLUSION

We have proposed the first method for BN structure learning that scales to large instances while respecting treewidth bounds and soft expert constraints. At the heart of our method is utilizing a MaxSAT encoding, applied locally, which demonstrates the flexibility of the SLIM framework.

We see several possibilities for improving the portion of satisfied expert constraints. An easy target is improving the Phase 1 heuristics to better handle the root bag construction, which a MaxSAT encoding could provide. Even more potential might be to adapt other heuristics like k-MAX [Scanagatta et al., 2018] or Elimination Trees [Benjumeda et al., 2019] for Phase 1.

The current implementation does not actively try to increase the satisfied constraints in Phase 2. Despite that, it was somewhat surprising for us to still see a significant increase in the number of satisfied constraints (see Table 5). This suggests a learning approach where we continuously check during Phase 2 whether any previously violated expert constraint is satisfied and if so, add it as a hard constraint to the Phase 2 engine. This way, Phase 2 could yield a monotonic increase in both the score and the number of satisfied constraints.

The local solver is essentially a CNF formula, and we have not exhausted its whole range of expressiveness with the constraints explored in this paper. Thus, another viable future direction could be to explore more sophisticated constraint types. Similarly, one can look into incorporating expert constraints into the heuristic learning algorithms for other probabilistic graphical models.

**Author Contributions**

S. Szeider and V. Peruvemba Ramaswamy worked on the initial concept and the write-up. V. Peruvemba Ramaswamy developed the software and performed the experiments.

**Acknowledgements**

The authors acknowledge the support by the FWF (projects P32441 and W1255) and by the WWTF (project ICT19-065).

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
