# OpenReview forum: "Learning Large Bayesian Networks with Expert Constraints"
_auai.org/UAI/2022/Conference — UAI 2022 Poster_

### Official Review · Reviewer_Mqqg · 2022-04-08

**Q2(1) Originality/Novelty:** 1
**Q2(2) Significance/Impact:** 2
**Q2(3) Correctness/Technical Quality:** 3
**Q2(6) Clarity Of Writing:** 3
**Q6 Overall Score:** 6
**Q8 Confidence In Your Score:** 4

**Q1 Summary And Contributions:**

This paper presents a new heuristic algorithm for learning structures of Bayesian networks. The heuristic allows to user bound the treewidth of the learned network as well as set constraints on different ancestor relations (i.e., whether or not there is an arc or a path between nodes u and v). The proposed method is a combination an existing greedy search method and an existing maxSAT-based method.

**Q2 Assessment Of The Paper:**

More detailed information regarding each of these aspects is given below:

**Q2(4) Quality Of Experiments (Optional):**

3: Good: The experimental evaluation is adequate, and the results convincingly support the main claims.

**Q2(5) Reproducibility:**

2: Fair: Key resources (e.g., proofs, code, data) are unavailable but key details (e.g., proof sketches, experimental setup) are sufficiently well-described for an expert to confidently reproduce the main results.

**Q3 Main Strengths:**

The proposed method seems to perform well in practice.

**Q4 Main Weakness:**

Limited novelty.

**Q5 Detailed Comments To The Authors:**

Section 2.1: The local score not consistent (both f_P(v) and f(v, P_D(v))

Section 2.2: The treewidth-bounded BN structure learning problem bounds the treewidth of the *moral graph*.

**Q7 Justification For Your Score:**

The overall evaluation of the paper is somewhat challenging. On one hand, the technical contributions are quite limited. In the end, the proposed method is just a straightforward combination of existing techniques. On the other hand, the combination seems to produce a method that performs well and is thus a valuable contribution to the field.

Overall, the most important quality for a heuristic is that it performs well in practice and thus I am leaning towards accept.

**Q9 Complying With Reviewing Instructions:**

1: Yes.

---

### Official Review · Reviewer_XNWi · 2022-04-10

**Q2(1) Originality/Novelty:** 2
**Q2(2) Significance/Impact:** 2
**Q2(3) Correctness/Technical Quality:** 3
**Q2(6) Clarity Of Writing:** 4
**Q6 Overall Score:** 7
**Q8 Confidence In Your Score:** 3

**Q1 Summary And Contributions:**

This research proposes and implements a framework for Bayes net structure learning that iteratively applies MaxSAT optimizations to a network originated with a k-greedy scoring algorithm. The key contributions are: the method is scalable to networks with thousands of nodes; treewidth can be constrained (to bound the network complexity); and user-defined constraints over arcs and ancestry (e.g., a node should -- or not -- be an ancestor of another) can be imposed.

**Q2 Assessment Of The Paper:**

More detailed information regarding each of these aspects is given below:

**Q2(4) Quality Of Experiments (Optional):**

3: Good: The experimental evaluation is adequate, and the results convincingly support the main claims.

**Q2(5) Reproducibility:**

3: Good: Key resources (e.g., proofs, code, data) are available and key details (e.g., proofs, experimental setup) are sufficiently well-described for competent researchers to confidently reproduce the main results.

**Q3 Main Strengths:**

It's a pretty well written paper. The ideas are articulated and presented in a logical sequence which helps with the understanding of the core concepts. Main contributions are made explicit and well separated from previous/existing work, and the claims about satisfiability of constraints (treewidth and user-defined arc/ancestry constraints) were clearly articulated and demonstrated with experiments. Limitations of current implementation are properly presented and plans for future improvements are also quickly discussed.

A technique that translates non-local ancestry constraints to local constraints are described in sufficient details, and this may suggest potential extensions to other applications/fields, such as "grounding" of a dynamic Bayes net models with constraints.

The source code and libraries provided in the supplementary material were especially helpful in following the procedures and algorithms discussed in the paper.

**Q4 Main Weakness:**

The handling of soft constraints does not seem to be clearly discussed. It's briefly presented in some paragraphs, not as a formal nor systematic method.

The supplementary material (source code, libraries) seems to lack some instrumentation/documentation (e.g., readme files are very simplistic), so it's slightly hard for understanding and (re)using.

Further analysis on scalability and asymptotic behavior (larger/complex networks) is desired, since one claimed contribution was on scalability.

Although this is common in papers that extend/combine previous works, this paper seems to use considerable space for repeating what's already stated in the citations, especially some definitions and their natural consequences. Although this resulted in a comprehensive paper, some of these spaces could have been used for illustrations to help prospective readers capture the main concepts (which are quite cumbersome/bulky at first glance).

**Q5 Detailed Comments To The Authors:**

Please, find below my two cents...

Please explain the acronym DAG (I read it as Directed Acyclic Graph) before or near its 1st use.

Similarly, please define/explain "path," preferably before or near its 1st use in the document (because it's not trivial to guess that "path" is limited only to "simple" directed path). This is mainly because table 1 in page 1 is already referring to "path", while the word "path" is only defined/explained later in page 3.

Is the name "Con-BN-SLIM" some acronym? If so, could you also provide its meaning in the paper?

Page 2 mentions "The empirical ﬁndings on our prototype are highly encouraging, providing avenues of further investigation."
It would be nice to explicitly list a few extension points in separate, as future works (together with the observations you made in the conclusion about possibilities for improving the framework).

In page 2, "local parameters" are mainly local/conditional probabilities of a node given its parents, or does it mean something else? Could you shortly clarify this in the paper (e.g., just add a short explanation between parenthesis).

Also in page 2, $f_P (v)$ is defined but never used within the paragraph, while $f(v,P)$ is used later in the paragraph. Does $f_P (v)$ =  $f(v,P)$ ? If so, please edit the paragraph.

Again in page 2, "$P_D(v) = { u ∈ V : (u, v) ∈ E }$" suggests for the 1st time that an edge in E is a set of 2 nodes in V.
Please explicitly define it (i.e., indicate that an edge is also subset of 2 nodes in G) in advance; especially because the same definition seems to be implicitly reused in Section 2.2 TREEWIDTH, while the 1st paragraph in section 2.1 just says "E is the set of arcs" (without indicating that it's also a pairs of nodes).

In section 2.3; can your framework accept negative ancestry constraints that impose disconnected nodes (i.e., subnets with no path between any nodes in different subnets)?
A slightly related question would be: can we specify constraints with first-order logic expressions? For example, "∀ x ∈ DAG ∃ y , (x⇝y) ∨ (y⇝x)"?

In page 3, the word "simple path" is only used here, so this sentence does not seem to add much value to the paper as a whole.
Perhaps it won't be necessary to explain a "simple" path once you define that a DAG is a graph with no directed cycles (which is equivalent to saying that a DAG has only simple paths)?
Also, page 3 mentions "directed" path for the first time. It might be a bit redundant because the definition in the same paragraph already implies that paths in a DAG are directed (because it's defined as a *sequence* of vertices such that there exists an arc **from** $v_i$ **to** $v_{i+1}$).

About sections 3.1-3.3: I totally agree about the importance of making the contributions explicit (i.e., keep existing/previous work separate from new extensions), but this paper already wrote them quite well in previous sections. I believe it would be easier for prospective readers to find sections 3.1-3.3 integrated (e.g., perhaps a single "Extended K-Greedy Algorithm" section) to avoid having to move the attention between subsections.
In addition, it would be nice to show more details of your modifications in a more precise/unambiguous language -- perhaps a listing with algorithmic pseudocodes?
In this case, you may need to shorten some sections to fit to the page limit. E.g., point to references instead of writing definitions, especially sections 2.1 and 2.2; and/or part of section 4.1 that restates what's already said in [Peruvemba Ramaswamy and Szeider, 2021a]).

In section 4.1, how do you guarantee that an iterative process starting from a heuristic solution D will reach a "maximum" score (instead of "local" maxima)?
Is this iterative process "greedy" (i.e., the next step will be always applied to the best $D^{new}$ known so far) or there is virtually something like a "backoff" mechanism (to return a few steps before or restart from another initial D -- to avoid local maxima)?

Also in section 4.1 (page 4), could you clarify (or add citations) about what's a "budget"?

In page 5, under "Positive ancestry constraints", you mention "Since the constraint is satisﬁed in D."
What would be the impact of soft constraints here, since Section 3.3 indicates that positive constraints might become mostly (if not all) soft constraints in practice? For instance, can your framework re-consider soft-constraints ignored in D?

Do you have plans to release the source code to the public/academy under some specific license (e.g., some open-source license), independently from the conference?


**Q7 Justification For Your Score:**

This is a clear and well organized paper.
Some of the core concepts, such as the translation of global constraints to local ones, and the iterative/adaptive use of MaxSAT algorithms to find network structures seem to be applicable in different domains as well, such as resolution of "abstract" BNs (e.g., dynamically & incrementally constructing "classic" BNs based on constraints on extended BNs with "higher" expressiveness).
The weaknesses mentioned in Q4 were minor compared to these strengths.

**Q9 Complying With Reviewing Instructions:**

1: Yes.

---

### Official Review · Reviewer_62js · 2022-04-13

**Q2(1) Originality/Novelty:** 2
**Q2(2) Significance/Impact:** 2
**Q2(3) Correctness/Technical Quality:** 3
**Q2(6) Clarity Of Writing:** 3
**Q6 Overall Score:** 7
**Q8 Confidence In Your Score:** 3

**Q1 Summary And Contributions:**

The paper modifies two existing BN structure learning algorithms such that they can handle constraints. These modified algorithms are then combined by feeding the output of the first algorithm (Con-k-greedy) as the initial input to the second one (Con-BN-Slim).

The modified algorithms and their combination allow for learning of bounded tree width BNs, the support of constraints, and the scaling to a larger number of random variables. The combination of these three has not been achieved before.

**Q2 Assessment Of The Paper:**

More detailed information regarding each of these aspects is given below:

**Q2(4) Quality Of Experiments (Optional):**

3: Good: The experimental evaluation is adequate, and the results convincingly support the main claims.

**Q2(5) Reproducibility:**

4: Excellent: Key resources (e.g., proofs, code, data) are available and key details (e.g., proof sketches, experimental setup) are comprehensively described for competent researchers to confidently and easily reproduce the main results.

**Q3 Main Strengths:**

The main strength of the paper lies in its simplicity. It takes two existing approaches that allow for learning the structure of large Bayesian networks of bounded tree-width and add the feature of incorporating in additionally constraints. The technical problem that the paper solves is how to incorporate these constraints in the existing algorithms.

**Q4 Main Weakness:**

I am not sure what I have learned from the paper. The way the paper reads it puts forward a method that glues together two algorithms into which constraints have been integrated. More specifically, what is the motivation of gluing SLIM algorithm to the k-greedy algorithm. This seems a little bit like an ad-hoc solution. This brings me also to a weakness of the paper’s writing (which is general however very clear). The paper reads a little bit like chronological report of what has been done, what failed and what succeeded.

Coming back to the ad-hoc combination of the two algorithms. Why was the Con-BN-Slim algorithm not experimentally evaluated without the initial structure being provided by the Con-k-greedy algorithm?

An other experiment that I think is missing is how the constrained BN learning algorithms perform under varying \eta, i.e. how do they perform when increasing the number of constraints, where in the extreme case no constraints are given.

**Q5 Detailed Comments To The Authors:**

- It might have been better to present the contributions of the paper as extending two existing BN learning algorithms with constraints. Then evaluate them and then explore the option of gluing them together.

- Also how does the BN-Slim algorithm (without constraints) perform when feeding it the k-greedy proposal BN (again without constraints).

-  you say that you were not able to integrate a constraint solver in the k-greedy algorithm. Why is that?

- Why do you use an approximate computation of the score. Don’t you have tractable computation of the likelihood for bounded tree width BNs?

- It might be good to not use the term “bucket” the way you do. In the BN literature bucket already has a specific meaning, cf. bucket elimination.

-Some more experiments actually showing that that other constrained BN learners do not scale would have been beneficial. Also how do the proposed algorithms fair against the competition with few RVs only.




**Q7 Justification For Your Score:**

The main idea is quite neat: taking existing algorithms and simply extending them with the ability to handle constraints, which lead me to give an overall positive score.

However, the main message of the paper is a bit unclear and I am not sure what the experiments say exactly and what exactly they evaluate (including constraints or combining the algorithms). This is why I give the paper a score more to the lower side.

**Q9 Complying With Reviewing Instructions:**

1: Yes.

---

### Official Review · Reviewer_zDCF · 2022-04-13

**Q2(1) Originality/Novelty:** 2
**Q2(2) Significance/Impact:** 2
**Q2(3) Correctness/Technical Quality:** 3
**Q2(6) Clarity Of Writing:** 3
**Q6 Overall Score:** 5
**Q8 Confidence In Your Score:** 4

**Q1 Summary And Contributions:**

The work proposes to discover Bayesian Networks (DAG) based on Data and prior knowledge. The approach extends and mixes two previous works on the topic and shows interesting experimental results on a set of classical problems.
The algorithm is a 2-phases approach. In a first phase, a “good” DAG is computed. However, it is shown that the proposed approach can only hardly respect all the constraints. In a second stage, this first model is refined.



**Q2 Assessment Of The Paper:**

More detailed information regarding each of these aspects is given below:

**Q2(4) Quality Of Experiments (Optional):**

3: Good: The experimental evaluation is adequate, and the results convincingly support the main claims.

**Q2(5) Reproducibility:**

4: Excellent: Key resources (e.g., proofs, code, data) are available and key details (e.g., proof sketches, experimental setup) are comprehensively described for competent researchers to confidently and easily reproduce the main results.

**Q3 Main Strengths:**

The overall goal is very interesting. Focus on fitting the data with expert constraints is really appealing. The constraints add the history of the data and allow to search for a correct BN representation of the data.
The paper is easy to follow and brings interesting and new results in learning BN.
The size of the problems handled by the approach is really a plus.


**Q4 Main Weakness:**

Even if the approach is new, it may be seen as a two small improvements over existing results. The improvement of the k-greedy algorithm is somehow straightforward. The section 3.2 adds a number of tricks that are not well motivated. Why not considering all positive constraints? Is there a reason for this choice?
The notion of treewidth is not clear in the designed flow. What if the DAG generated in the first phase is highly connected?
I understand the idea of the two-phase algorithm, but why not designing the overall approach as a single algorithm? It is clear that, once the first phase has found a solution, the second phase can only work on improving it.


**Q5 Detailed Comments To The Authors:**

The paper is interesting to read and fairly reports the limits of the approach.
The notion of heuristics is not clear in the paper. In AI, this is generally a static function. Not an algorithm. May be a definition would fix this.

The random sampling of a partial ordering may introduce some bias and it is may be not obvious to sample randomly

The Figure 2 tends to try an approach where the second phase is quickly launched, because the reported improvements are way above the improvements observed by the k-greedy search. Why splitting the time in 2 times 30 minutes?

The titles of the sections / subsections are not always well chosen.

Section 3.2 is really problematic. The choices made there should be supported by experimental evidences (how would the experimental results evolve if the hypothesis are changed in sec 3.2?).
It is also claimed that it was not possible to integrate a constraint-sensitive local solver… This should be clarified because then, the reader does not understand what is really done. How for instance to “force” the solver to consider a given hard constraint?

The first phase may generate a DAG that can only be extended to local minima by the second phase. It may be interesting to report the sensitivity of the algorithm by running multiple times the same problem and studying it too.

The paper claims to give the source code of the CON-BN-SLIM solver but, as far as I checked, only the compiled bytecode is given in a java .jar archive. However, it is clear that the binaries are sufficient to reproduce the results (which is good).



**Q7 Justification For Your Score:**

The paper improves the state of the art in learning BN by adressing two main problems : (1) taking into account the constraints given by experts and (2) handling very large BN.
However, the paper still have some weakness and the positive feeling must be conter balanced by the weak points that I raised above and in the detailed comments.

**Q9 Complying With Reviewing Instructions:**

1: Yes.

---

### Decision · Program_Chairs · 2022-05-15

**Decision:**

Accept (Poster)

**Comment:**

Meta Review: This paper introduces a new structure learning algorithm for Bayesian networks that allows one to restrict the search space to bounded treewidths and user-defined arc/ancestry constraints. The main strength is the novel and simple way in which search space restrictions are incorporated into an structure learning algorithm. The main weakness is that the experimental results could be improved.